# Dynamic Implant Surgery—An Accurate Alternative to Stereolithographic Guides—Systematic Review and Meta-Analysis

**DOI:** 10.3390/dj11060150

**Published:** 2023-06-08

**Authors:** Jordi Marques-Guasch, Anna Bofarull-Ballús, Maria Giralt-Hernando, Federico Hernández-Alfaro, Jordi Gargallo-Albiol

**Affiliations:** 1Department of Oral and Maxillofacial Surgery, Universitat Internacional de Catalunya, Josep Trueta s/n, Sant Cugat del Vallès, 08195 Barcelona, Spain; 2Department of Periodontics and Oral Medicine, University of Michigan School of Dentistry, Ann Arbor, MI 48109, USA

**Keywords:** surgical navigation systems, dynamic surgery, computer-aided surgery, computer-assisted surgery, dental implant, accuracy, meta-analysis

## Abstract

(1) Background: Dynamic guided surgery is a computer-guided freehand technology that allows highly accurate procedures to be carried out in real time through motion-tracking instruments. The aim of this research was to compare the accuracy between dynamic guided surgery (DGS) and alternative implant guidance methods, namely, static guided surgery (SGS) and freehand (FH). (2) Methods: Searches were conducted in the Cochrane and Medline databases to identify randomized controlled clinical trials (RCTs) and prospective and retrospective case series and to answer the following focused question: “What implant guidance tool is more accurate and secure with regard to implant placement surgery?” The implant deviation coefficient was calculated for four different parameters: coronal and apical horizontal, angular, and vertical deviations. Statistical significance was set at a *p*-value of 0.05 following application of the eligibility criteria. (3) Results: Twenty-five publications were included in this systematic review. The results show a non-significant weighted mean difference (WMD) between the DGS and the SGS in all of the assessed parameters: coronal (n = 4 WMD = 0.02 mm; *p* = 0.903), angular (n = 4 WMD = −0.62°; *p* = 0.085), and apical (n = 3 WMD = 0.08 mm; *p* = 0.401). In terms of vertical deviation, not enough data were available for a meta-analysis. However, no significant differences were found among the techniques (*p* = 0.820). The WMD between DGS and FH demonstrated significant differences favoring DGS in three parameters as follows: coronal (n = 3 WMD = −0.66 mm; *p* =< 0.001), angular (n = 3 WMD = −3.52°; *p* < 0.001), and apical (n = 2 WMD = −0.73 mm; *p* =< 0.001). No WMD was observed regarding the vertical deviation analysis, but significant differences were seen among the different techniques (*p* = 0.038). (4) Conclusions: DGS is a valid alternative treatment achieving similar accuracy to SGS. DGS is also more accurate, secure, and precise than the FH method when transferring the presurgical virtual implant plan to the patient.

## 1. Introduction

Dental implants are used in many different clinical scenarios and have been shown to achieve high survival and success rates. However, implant complications are often related to incorrect positioning [1,2,3].

The development of implant planning software has considerably improved three-dimensional implant positioning regarding both the anatomical and prosthetic requirements [4,5].

The introduction of guided surgery has allowed clinicians to relate the anatomical information obtained from cone-beam computed tomography systems (CBCTs) [6,7] to prosthetic planning, determining the implants’ positions so that they meet the requirements for a predictable, aesthetic, and functional outcome, while respecting the anatomical structures [7,8].

When the exact implant location is decided presurgically and the anatomical characteristics are favorable, it is possible to perform flapless surgery and thus preserve blood supply and reduce both surgical time and the patient’s postoperative discomfort [7,8]. Static guided surgery (SGS) uses either computer-aided design/computer-assisted manufacturing (CAD/CAM) or a handmade dental laboratory guide for placing the implants. The surgical splint guides bone drilling and implant placement at the predetermined location, angulation, and depth [9,10]. The surgical template may be tooth-, mucosa-, or bone-supported, or even a combination of these [11]. Although SGS is regarded as a highly accurate method, it does suffer from certain limitations [12,13]. SGS requires a wide mouth-opening range to introduce specific instruments into the oral cavity, especially when treating posterior regions; the guide interferes with irrigation during osteotomy; it may fracture during surgery; and last-minute planning modifications are not possible, among other issues [9,14,15].

Dynamic guided surgery (DGS) offers an alternative guidance method, which systemizes computer-assisted navigation in real time [16,17]. A screen displays the current position of the handpiece in relation to various CBCT images, which, in turn, are related to the prosthetic plan. The most common DGS uses optical cameras to track the handpiece over the CBCT data and guide the surgeon throughout the realization of a predetermined virtual plan [18,19]. No surgical templates are needed, and intraoperative modifications of the initial treatment plan can be introduced at any moment, if required [20]. With no surgical template and no need for specific instrumentation, limited mouth opening is no longer a problem [21]. However, the technique demands a considerable learning curve and higher economic costs, and provides a limited vision of the surgical field, all of which are factors for improvement in future developments [15,22,23].

Although DGS appears to provide a more precise guidance method than the conventional freehand (FH) method, deviations between virtual planning and actual clinical implant positions have been reported [24]. Only limited data on the accuracy of the DGS system are available, and the literature published to date is very diverse.

For these reasons, the aim of this systematic review was to determine the precision of the dynamic guided system (DGS) compared with the static guided system (SGS) and freehand (FH) approaches, and to describe the current state of the art.

## 2. Materials and Methods

We aimed to answer the question “In partially edentulous patients with the need for dental implants, what implant guidance tool is more accurate and secure for implant placement surgery?” by following the PRISMA (Preferred Reporting Items for Systematic Review and Meta-Analyses) guidelines, which consist of a 27-item checklist [25] (referring to the title, abstract, introduction, methods, results, discussion, and funding) and the patient, intervention, comparison, and outcomes (PICO) method, as follows. The patients (P) were those receiving dental implant placement surgery, the intervention (I) was implant placement using a dynamic navigation guidance method, the comparison © was with static guidance or freehand implant placement, and the outcome (O) was the accuracy of implant placement between presurgical planning and postsurgery outcomes in terms of horizontal, vertical, and angular deviation.

### 2.1. Search Strategy

Systematic searches were conducted in the Cochrane Central Register of Controlled Trials, the Cochrane Oral health group, and the National Library of Medicine (Medline via PubMed). Combinations of different thesaurus terms for indexing the articles were the medical subject headings (MeSHs) and EMTREE terms used as search terms as follows: (“partial edentulism,” OR “edentulous jaw” OR “edentulous jaws” OR “jaws, edentulous” OR “jaw, edentulous, partially”) AND (“implant, dental” OR “implants, dental,” OR “dental implant”), AND (“computer-assisted surgeries” OR “surgeries, computer-assisted” OR “surgery, computer assisted” OR “computer-assisted surgery” OR “computer assisted surgery” OR “computer-aided surgery” OR “computer aided surgery” OR “computer-aided surgeries” OR “surgeries, computer-aided” OR “surgery, computer-aided” OR “surgery, image-guided” OR “image-guided surgeries” OR “surgeries, image-guided” OR “surgery, image guided” OR “image-guided surgery” OR “image guided surgery” OR “surgical navigation”, OR “navigation, surgical”). The same strategy was used in the case of the Cochrane Library, since it also uses MeSH terms. Gray literature from the Google Scholar Beta database was also searched to retrieve studies published in journals not indexed in the major databases. All duplicates from the systematic searches were subsequently removed.

A manual search was performed of the abstracts and references of articles from the following Scopus-indexed dental implant journals: *Journal of Oral Implantology*, *Clinical Oral Implants Research*, *International Journal of Oral and Maxillofacial Surgery*, *Journal of Dental Implant Research*, *Clinical Implant Dentistry and Related Research*, *International Journal of Implant Dentistry*, and *International Journal of Clinical Implant Dentistry*.

### 2.2. Study Selection

The electronic search and the selection of studies was conducted by two authors (J.M.G. and A.B.B.) to avoid subjectivity. The systematic electronic, manual, and gray literature searches were applied between January 2000 and December 2021. The studies that fulfilled the inclusion criteria were retrieved for full-text reading.

The inclusion criteria were studies reporting the accuracy of implant positioning, at least one deviation type (horizontal implant platform, horizontal implant apex, implant angulation, and the implant platform on the vertical plane), and the accuracy of implant placement determined by differences detected by superposition of the radiological 3D presurgical implant plan and a postoperative CBCT.

The following exclusion criteria were applied: (1) articles written in languages other than English, (2) studies published before the year 2000, (3) review articles or in vitro studies (including studies of animal or cadaver models), (4) case series with fewer than five patients, (5) articles with missing information that could not be deduced, such as the total number of patients and/or implants (the authors were contacted to supply missing information; failure to respond resulted in exclusion), (6) evaluation methods that were different from the one described above, and (7) studies in which the implants were placed in fully edentulous patients.

Discrepancies were resolved by discussion and consensus. If any discrepancy could not be resolved, a third reviewer (J.G.A.) was consulted. The level of interrater agreement between authors was assessed using Cohen’s kappa coefficient (κ).

### 2.3. Data Extraction

Demographic, qualitative, and quantitative data were compiled from the included studies as follows: title; author; year of publication; study design; number of patients; number of implants; guidance method used; number of patients and implants subdivided into subgroups; maxillary or mandibular implants; anterior or posterior location; immediate, early, or delayed loading; survival rate; dynamic guidance system used; surgical complications; prosthetic complications; accuracy of the implant at the implant platform; accuracy of the implant at the implant apex (apical), angular accuracy, implant accuracy at the implant platform in the vertical plane (apico-coronal); and the accuracy of the evaluation method.

### 2.4. Quality Assessment: Risk of Bias in Individual Studies

Two independent reviewers (J.M.G. and A.B.B.) evaluated the risk of bias. If any discrepancies were found, they were resolved by discussion until a consensus was reached.

The quality assessment of the included randomized clinical trials (RCTs) was performed according to the *Cochrane Handbook for Systematic Reviews of Interventions* (Version 5.1.0, updated March 2011) [26] and the CONSORT statement [25]. Seven parameters were assessed for each study: random sequence generator, concealment of the allocation, blinding of participants and personnel, blinding of the assessment of the outcome, incomplete outcome data, selective outcome reporting, and other sources of bias.

The risk of bias for each RCT was considered to be moderate if one of the analyzed parameters was not clear enough to be categorized as having a low risk of bias. In the event that the study had two or more parameters with a moderate risk of bias, the study was categorized as having a high risk.

The Newcastle–Ottawa Scale (NOS) [27] was used for cohort studies analyzing three categories: selection of the study groups, and comparability between the groups and outcomes. The Robins-I scale (-) [28] was applied to case series without a control group. In this case, the categories analyzed were before the intervention, at intervention, and after the intervention.

### 2.5. Definitions of the Outcomes

The different types of deviation from preplanned implant positions analyzed in the review were those described by Schneider et al. [29]. In all the parameters evaluated, the accuracy or deviation of the implant referred to the discrepancy between the planned implant and the actual clinical position of the implant after placement.

-Deviation of the implant platform refers to any discrepancy at the most coronal part of the implant (connection) in two spatial dimensions: buccal–lingual and/or mesial–distal direction.-Apical deviation of the implant refers to any discrepancy at the most apical part of the implant (apex or tip) in two spatial dimensions: buccal–lingual and/or mesial–distal direction.-Vertical deviation of the implant refers to any apico-coronal discrepancy measured at the most coronal part of the implant (platform). Angular deviation refers to any discrepancy (expressed as degrees) of the whole implant body.

### 2.6. Statistical Analysis

The software used for statistical analysis of the data extracted from the articles was R 3.5.1 (R Core Team, 2020, RStudio, PBC, Boston, MA, USA). We investigated four primary outcomes: (1) horizontal deviation at the implant platform (mm), (2) horizontal deviation at the implant’s apex (apical) (mm), (3) angular accuracy (°), and (4) accuracy in the vertical or apico-coronal plane (mm). The significance level was set at 5% (*p* = 0.05). The following analyses/calculations were performed.

In some studies, the information is presented by subgroups (e.g., platform deviation was expressed as two values, one corresponding to anterior implants and another to posterior implants). In these cases, a weighted mean (WM) and pooled standard deviations were applied to obtain a single measurement. If the subgroups corresponded to related measurements (e.g., the deviation was expressed as two values, one corresponding to deviation produced on the buccal side and another on the lingual side), a high correlation was considered (*p* = 0.7) to obtain the overall standard deviation.

Two groups of meta-analyses were performed. The first compared different techniques for guiding the placement of the implant when all available subgroups regardless of the design of the study were included. The second estimated the global effect measure of all subgroups corresponding to a specific guidance technique (whether it came from one-arm or multiple-arm studies). The estimation was carried out for RCTs, prospective studies (PSs), and retrospective studies (RSs) separately and all three together.

To compare the studies, the mean values of the primary outcomes were directly pooled and analyzed with the weighted mean differences (WMDs) and 95% confidence intervals (CIs). Random effect models [30] were used. Additionally, a meta-regression to assess the differences in the deviations obtained for the different techniques was also applied.

Heterogeneity was tested by means of the I^2^ index (the percentage of total variability due to heterogeneity) and the null statistical test (Q). Consistency across the results was explored via Galbraith plots.

To assess the studies’ selection bias, the Egger test was applied, and it is represented by funnel plots.

## 3. Results

### 3.1. Study Selection

The three-phase flowchart (identification, screening, and inclusion) (Figure 1) shows each step of the systematic search, confirming the thoroughness of the screening process [26].

The main electronic search yielded a total of 424 articles. After we had assessed the titles and abstracts, 143 complete articles were selected for screening and 78 studies were excluded. Of 65 studies assessed for eligibility, 38 were excluded because they did not meet the inclusion criteria. During the full-text analysis, two studies [21,31] were excluded because they did not specify how many implants were placed, and three additional studies were excluded [32,33,34] because they reported the accuracy in a format that meant the relevant data could not be used (such as coordinates of the x, y, and z axes). Consequently, 22 investigations were ultimately included in the meta-analysis. The coefficient of interrater agreement was κ = 0.856 (95% CI: 0.773 to 1) for study selection.

### 3.2. Qualitative Analyses

Out of 22 selected investigations, 8 were RCTs, 10 were prospective case series, and 4 were retrospective case series. All the selected studies aimed to evaluate the accuracy of implant placement via the DGS, SGS, and/or FH method comparing the presurgical implant plan and the postsurgical CBCTs.

All the controlled studies compared two arms, whereas only one prospective study [31] evaluated the three different guidance methods, so data for each experimental group were analyzed independently. A split-mouth design was used in two RCTs [32,33].

For the present systematic review, we pooled data from 1192 patients with a total of 1880 implants. Among the included investigations, six studies segmented the results into subgroups [21,34,35,36,37,38] (e.g., single tooth vs. free-end, mesial–distal deviation vs. buccal–lingual deviation, the use of four teeth or fewer to support the guide vs. the use of five or more teeth).

Regarding flap reflection, six studies [36,37,38,39,40,41] combined the flap and flapless approaches, while five studies placed all the implants after raising a flap [32,33,42,43,44]. Four papers did not specify what approach was used [34,45,46,47]. The year of publication of the investigations ranged between 2008 and 2020. An overview of the studies’ characteristics can be found in Table 1, and the accuracy results of each individual study and type of intervention are shown in Table 2.

### 3.3. Quantitative Analyses

#### Platform Deviation

The results from the meta-analysis show a non-significant weighted mean difference between DGS and SGS of 0.02 mm (95% CI (−0.27, 0.31) *p* = 0.903). However, the comparison between DGS and the FH method found a weighted mean difference of −0.66 mm (95% CI (−0.74 to −0.59) *p* =< 0.001), indicating a significant difference between these techniques (Table 3). The heterogeneity of these two analyses was high for the comparison between DGS and SGS (I^2^ = 86%, *p* < 0.001) and moderate for the comparison between DGS and FH (I^2^ = 53%, *p* = 0.177). The results are shown in Figure 2 and Figure 3.

The global effect for each technique was analyzed individually. This was 0.86 mm (95% CI (0.60, 1.13) *p* =< 0.001) for dynamic guided surgery; 1.03 mm (95% CI (0.66, 1.40) *p* =< 0.001) for static guided surgery, and 1.61 mm (95% CI (1.45, 1.76) *p* =< 0.001) for the freehand technique. Figure 4 illustrates these findings in forest plots according to the guidance method.

### 3.4. Angular Deviation 

The meta-analysis showed a non-significant weighted mean difference between DGS and SGS of −0.62° (95% CI (−1.33, 0.09), *p* = 0.085), indicating considerably high heterogeneity (I^2^ = 79%, *p* < 0.001). However, a significant weighted mean difference of −3.52° was found between DGS and the FH method (95% CI (−4.69°–2.35°), *p* < 0.001) (Table 4), together with high heterogeneity (I^2^ = 98%, p = 0.951). These findings are summarized in forest and funnel plots (Figure 5 and Figure 6). The accuracy found for each guidance system was 3.40° (95% CI (2.54, 4.27), *p* =< 0.001), 3.44° (95% CI (1.97, 5.17), *p* =< 0.001), and 6.99° (95% CI (5.36, 8.63) *p* =< 0.001) for the DGS, SGS, and FH techniques, respectively.

### 3.5. Apical Deviation

The comparison between DGS and SGS yielded a non-significant weighted mean difference of 0.08 mm (95% CI (−0.11, 0.26), *p* = 0.401) and demonstrated total homogeneity (I^2^ = 0% *p* = 0.521). In contrast, the comparison between DGS and the FH technique revealed a significant weighted mean difference of −0.73 mm (95% CI (−0.88, −0.59), *p* =< 0.001) Table 5 associated with moderate homogeneity (I^2^ = 0% *p* = 0.521). The results are presented in Figure 7 and Figure 8.

The weighted mean accuracy found for the dynamic guided surgery technique was 1.27 mm (95% CI (0.98, 1.57), *p* =< 0.001) while that for static guidance was 1.14 (95% (0.78, 1.49), *p* =< 0.001) and that for the freehand approach was 2.33 (95% CI (2.10, 2.56), *p* =< 0.001). The results are illustrated in Figure 9.

### 3.6. Apico-Coronal Deviation

Because no studies compared DGS with SGS and only one compared DGS and FH [21], no meta-analysis of apico-coronal deviation could be performed.

However, the accuracy found for each individual technique was 0.61 mm (95% CI (0.45, 0.76) *p* =< 0.001) for DGS, 0.64 mm (95% CI (0.42, 0.86) *p* =< 0.001) for SGS, and 1.04 mm (95% CI (0.71, 1.36) *p* =< 0.001) for the FH technique. The heterogeneity was 96.9%, 93.1%, and 89.4%, respectively.

No significant differences between DGS and SGS (*p* = 0.820) were found. However, a significant difference (*p* = 0.038) was found between DGS and the FH method. Forest plots according to the type of technique are shown in Figure 10.

### 3.7. Risk of Bias Assessment

Cohort studies showed a risk of bias of between 8 and 9 out of a possible total score of 9. The domains that exhibited the greatest risk were comparability and selection (Appendix A).

In the case of RCTs, the domains showing the greatest risk were blinding of the participants and personnel, and the random sequence generator (Appendix A). The Robins-I scale showed that the highest risk of bias was before the intervention and during the intervention (Appendix A).

## 4. Discussion

The findings of this meta-analysis show similar accuracy between DGS and SGS. However, the FH method revealed a much higher level of inaccuracy. Apart from that, deviations between the planned and final clinical positions of the implant were found in all the studies reviewed, involving all the surgical navigation approaches [10,16,21,22,24,39,41,52,54,55,56,57,58,59,60,61].

Several authors have described different factors that might have some impact on the overall accuracy of implant positioning when DGS is used [24,52,62]: (1) misadjustment, movement, or loss of the guide or the jaw attachment, or the patient moving during surgery or the CT scan; (2) the system was susceptible to hand tremors, which can be a cause of deviation; (3) the accuracy may have been influenced by non-operative factors, such as obtaining the CT-CBCT data; (4) target registration errors (TREs), which refer to imperfect coordination of the tracking system between the drill tip and the corresponding point on the CBCT image after registration; and (5) optical tracking being affected by the noise produced by mechanical, thermal, or optical changes since the last time the system was calibrated. Overall, one factor frequently named in the literature as a source of error was the operator’s experience and the steep learning curve that dynamic implant navigation demands, which could be the reason why this system has not yet been widely accepted in clinical practice [21,22,24]. In inexperienced surgeons, the learning curve tends to flatten after placing around 20 implants [31,55,63]. Even though operational experience can improve proficiency, the learning curve does not improve beyond a certain level due to the system’s slight error value, a factor that must be taken into consideration [22].

Comparisons between SGS, DGS, and FH have been already made by several authors [10,39,57,59]. Somogyi-Ganss et al. [57] compared the static guided surgery method using three different types of stereolithographic guides with the dynamic guided system and a non-stereolithographic laboratory guide, finding the same level of accuracy in the dynamic and the static methods. Nevertheless, DGS had significantly greater precision than the laboratory acrylic template. In agreement with these results, Kaewsiri et al. [39] conducted an RCT that compared DGS with stereolithographic guides, but none of the parameters evaluated presented statistically significant differences between the techniques. However, Kan Sang-Hoon et al. [10] found greater precision with SGS for all the parameters investigated, except for the vertical plane in the canine region. Additionally, several systematic reviews and meta-analyses of DGS have reported very similar accuracy values to the present review [8,29,55,64,65,66,67]. Block et al. [21,55] made an interesting comparison between the accuracy of DGS and the FH method, finding considerably greater precision with DGS for all the parameters analyzed. The no statistical difference between DGS and SGS might also be explained because of their similarity in terms of implant planning and the surgical execution in similar clinical scenarios due to the inclusion criteria of this study.

Despite the DGS and the SGS have proven to be accurate implant systems, a certain safety margin for the anatomical structures must be respected because there is always a deviation in all dimensions of the space.

Notwithstanding the strict screening process used in this review, some limitations could have produced a biased outcome. Firstly, we included non-randomized studies with considerable differences in their sample sizes. Secondly, generally high heterogeneity was observed. arising from the influence of multiple variations among articles. such as the operators’ experience or drilling protocols. However, it was not possible to segregate factors to perform a more homogeneous analysis. Although the interpretation of the data obtained was problematic, DGS was found to achieve acceptable outcomes in terms of the accuracy of implant positioning in comparison with presurgical planning. More randomized clinical trials with standardized protocols that report long-term survival and success rates and provide better descriptions of the implantation areas are necessary to order to reduce the heterogeneity among studies.

It may be concluded that DGS is an accurate implant placement system that provides a level of precision between the 3D presurgical planning and the actual clinical placement of around 1 mm vertically and horizontally and 3.6° when considering the angular deviation. However, the technique showed high variability, and we suggest that practitioners be very cautious when planning and executing this technique near anatomical structures, especially during the learning curve. In comparison with other guidance methods, it appears that DGS achieves better accuracy than the freehand method and similar precision to static guided surgery using stereolithographic guides. Further studies are needed to better assess the accuracy of DGS with less heterogeneity.

## Figures and Tables

**Figure 1 dentistry-11-00150-f001:**
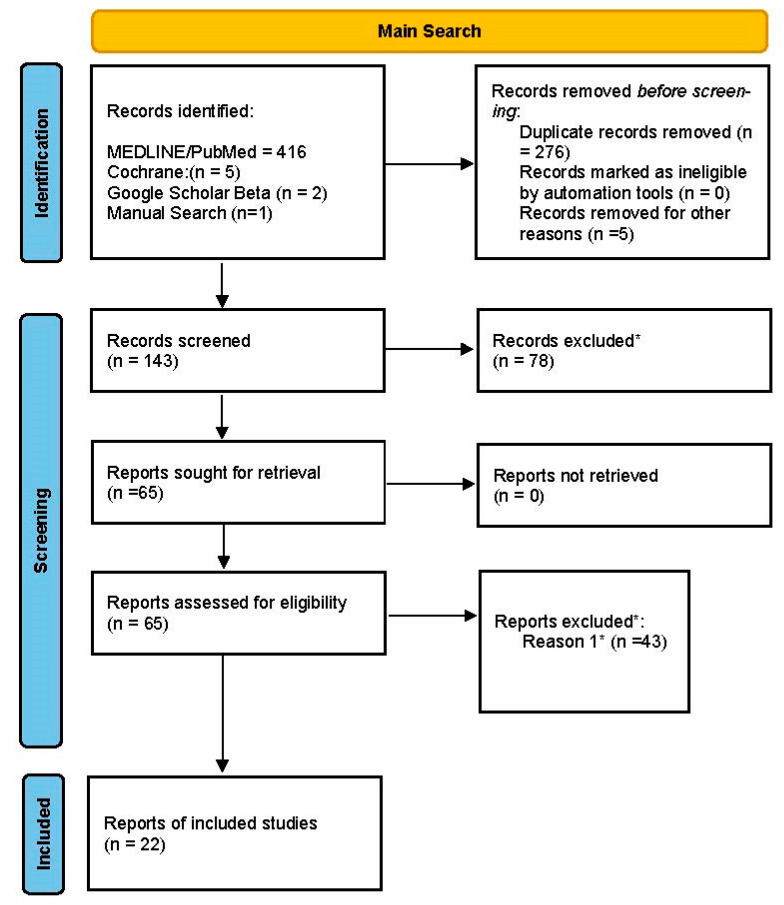
PRISMA flowchart depicting the article selection process.

**Figure 2 dentistry-11-00150-f002:**
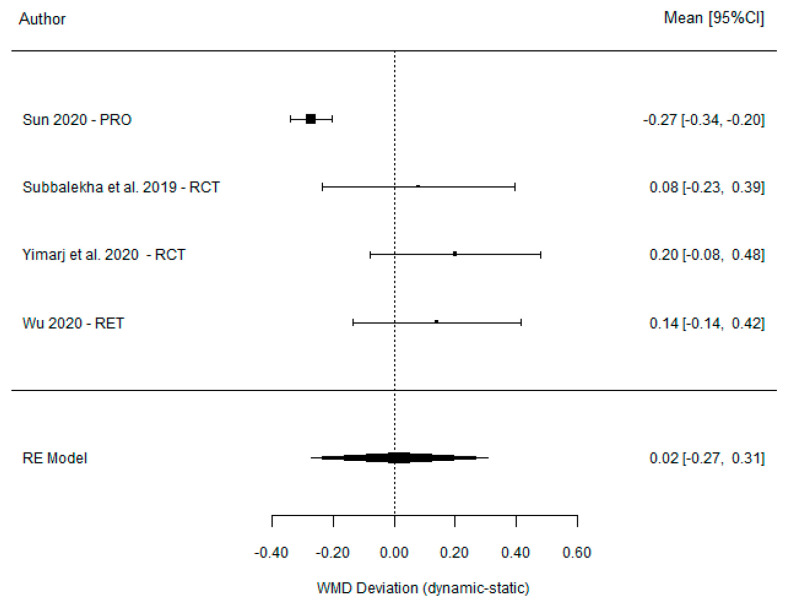
Forest plot illustration of the platform deviation (comparison between DGS and SGS).

**Figure 3 dentistry-11-00150-f003:**
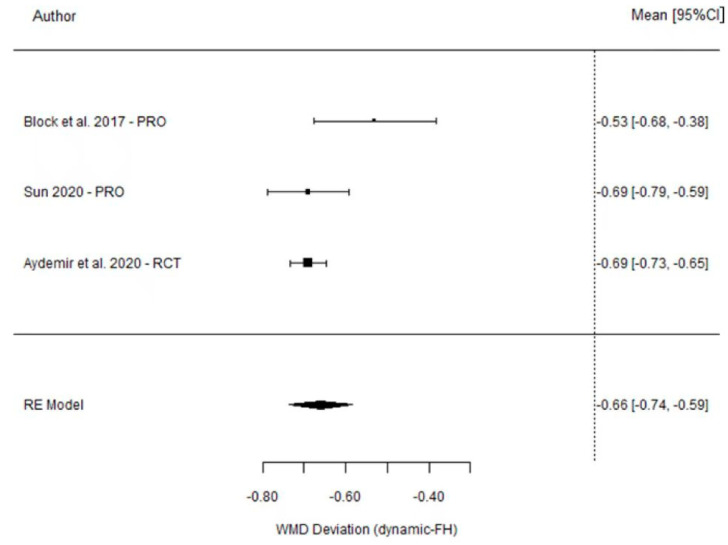
Forest plot illustration of the platform deviation (comparison between DGS and FH).

**Figure 4 dentistry-11-00150-f004:**
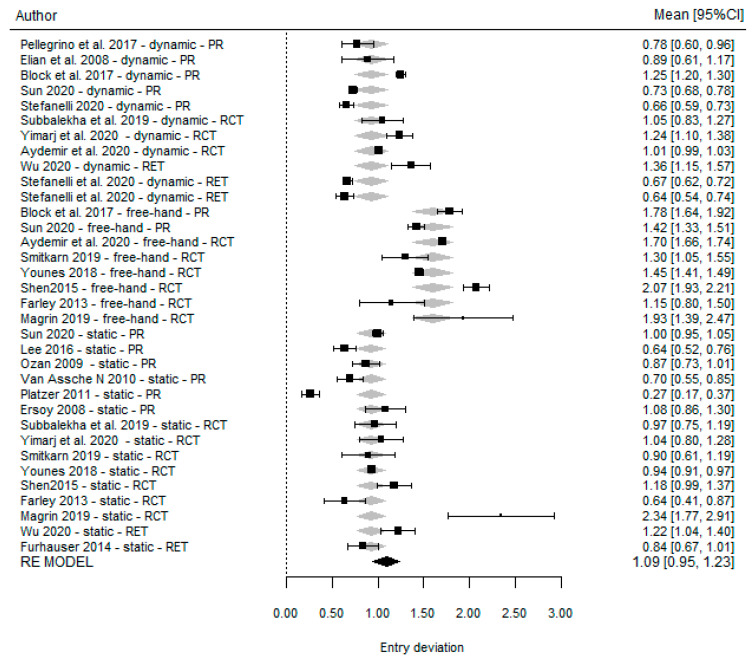
Forest plots for all studies reporting platform accuracy, according to the type of technique.

**Figure 5 dentistry-11-00150-f005:**
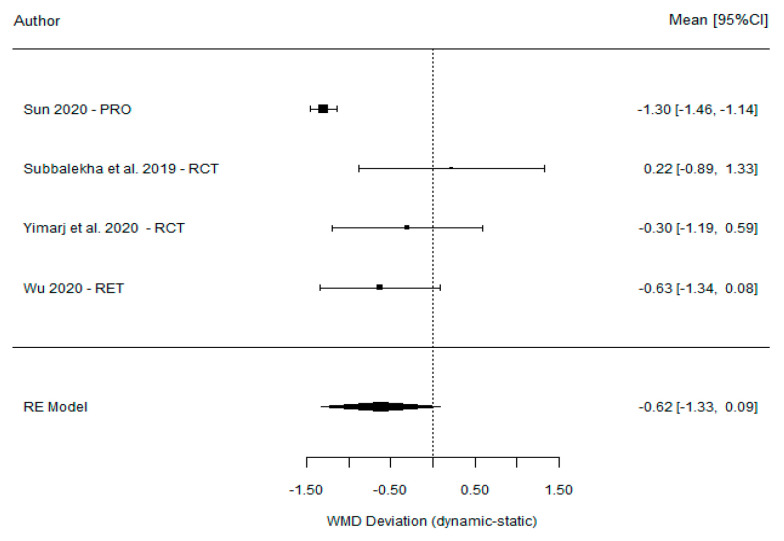
Forest plot illustration of the angular deviation (comparison between DGS and SGS).

**Figure 6 dentistry-11-00150-f006:**
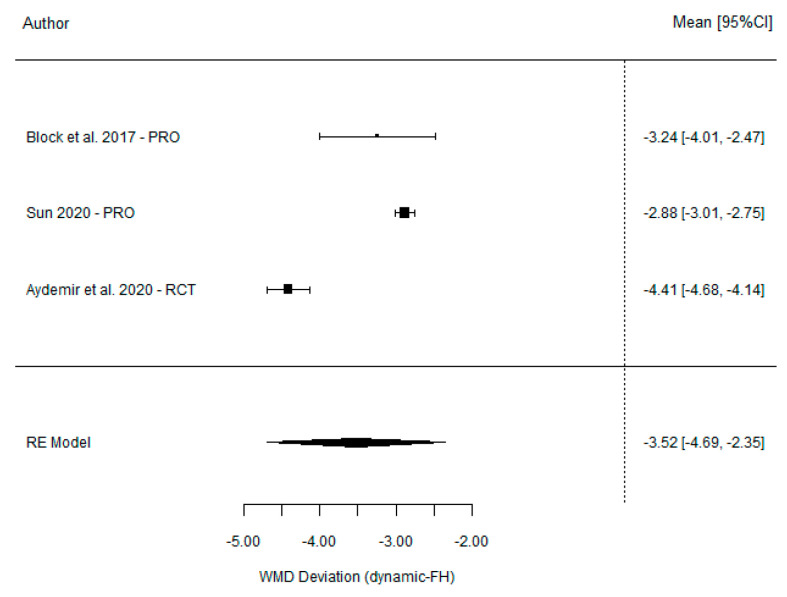
Forest plot illustration of the angular deviation (comparison between DGS and FH).

**Figure 7 dentistry-11-00150-f007:**
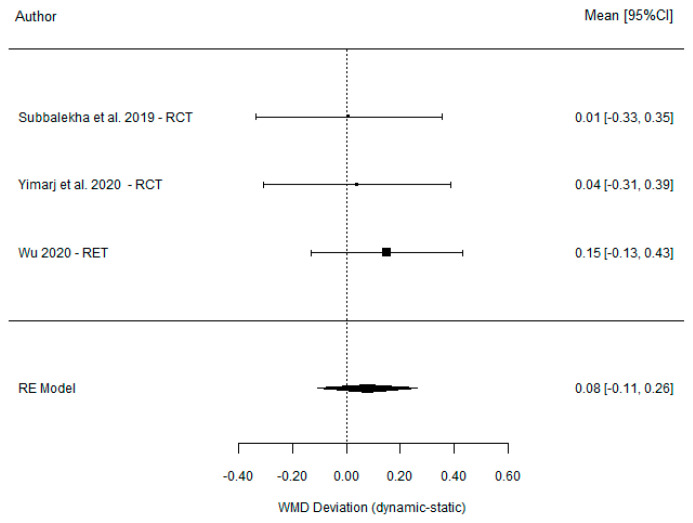
Forest plot illustration of the apical deviation (comparison between DGS and SGS).

**Figure 8 dentistry-11-00150-f008:**
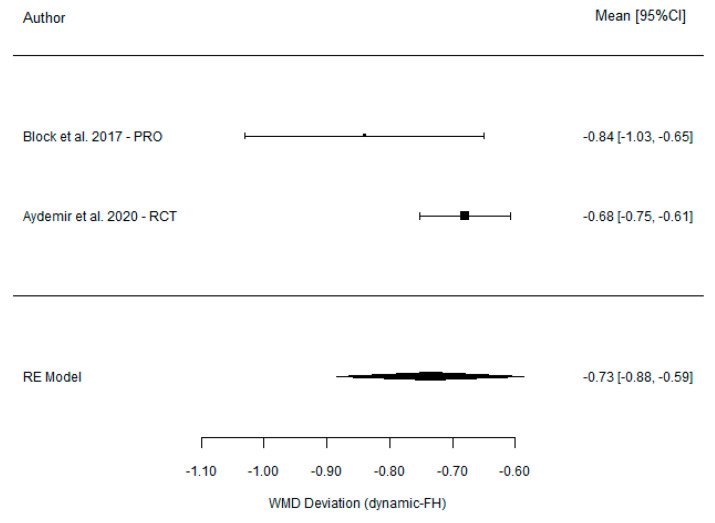
Forest plot illustration of the apical deviation (comparison between DGS and FH).

**Figure 9 dentistry-11-00150-f009:**
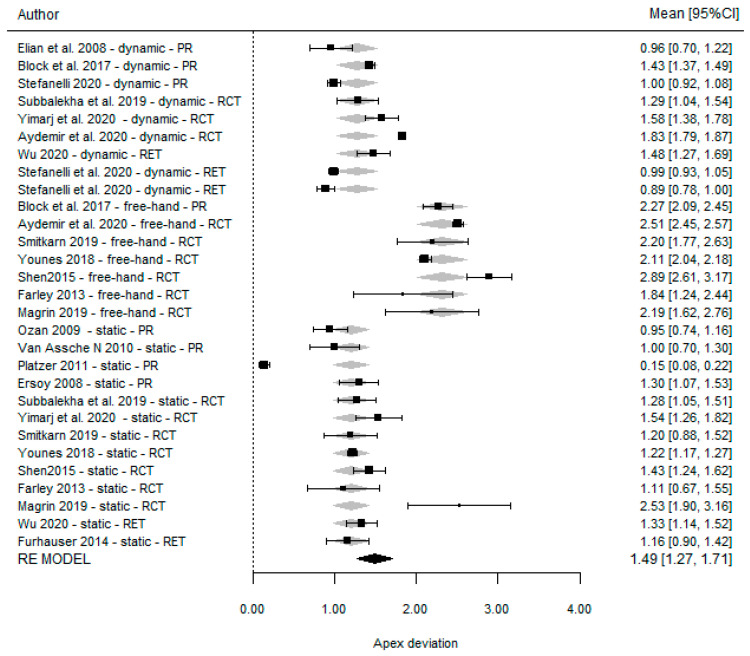
Forest plots for all studies reporting apical accuracy according to the type of technique.

**Figure 10 dentistry-11-00150-f010:**
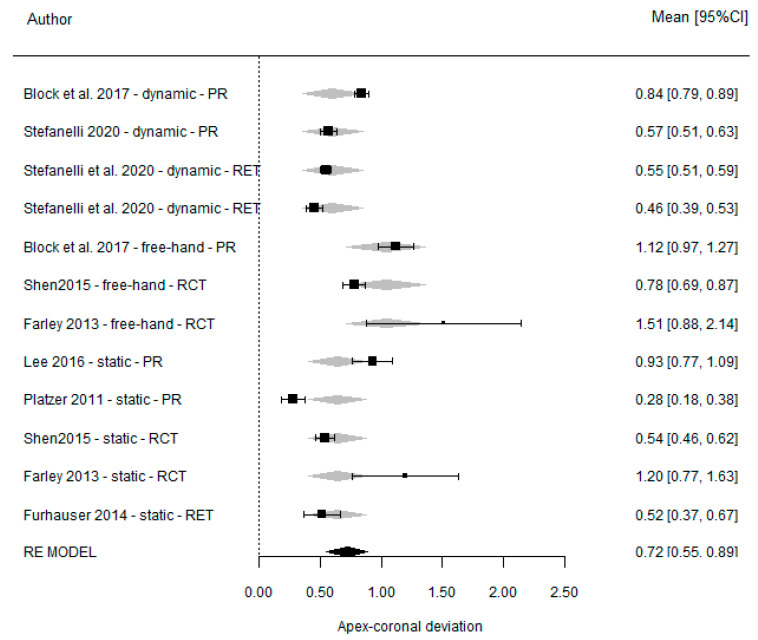
Forest plots for all studies reporting vertical accuracy, according to the type of technique.

**Table 1 dentistry-11-00150-t001:** General characteristics of the included studies.

Author (Year)	Study Design	Groups Analyzed	No. of Patients	No. of Implants	Location	Flap/Flapless	Dynamic Guided Navigation System	Surgical Complications
Dechawat Kaewsiri et al. [42] (2019)	RCT	DGS vs. SGS	30	30	Max and mand	Flap and flapless	Iris-100 software (EPED Inc., Taiwan)	NR
Paweena Yimarj et al. [47] (2020)	RCT	DGS vs. SGS	30	60	NS	NS	Iris-100 software (EPED Inc., Taiwan)	NR
Aktolun Aydemir et al. [37] (2020)	RCT	DGS vs. FH	32	86	Max	Flap	Navident (ClaroNav Inc., Toronto, ON, Canada)	Not satisfactory stability of the radiopaque stent in 2 patients
Palita Smitkarn et al. [45] (2019)	RCT	SGS vs. FH	52	60	Max and mand	Flap	CoDiagnostics	
Faris Younes et al. [41] (2018)	RCT	SGS vs. FH	33	71	Max	Flap and flapless	Simplant	NR
Pei Shen et al. [48] (2015)	RCT	SGS vs. FH	60	109	NS	Flapless	Simplant	NR
Farley et al. [36] (2013)	RCT	SGS vs. FH	10	20	Max and mand	Flap	Implant master software (iDent Imaging)	NR
Magrin et al. [43] (2019)	RCT	SGS vs. FH	16	24	Mand	Flap and flapless	DentalSlice, Bioparts	4 implants lacked osseointegration
Pellegrino et al. [44] (2017)	CS (prosp)	DGS	5	5	Max and mand	Flap and flapless	Navident (ClaroNav Inc., Toronto, ON, Canada)	NR
Elian et al. [49] (2008)	CS (prosp)	DGS	6	14	Max and Mand	Flapless	Software DenX Advanced Dental Systems, (Moshav Ora, Israel).	NR
Block et al. [21]	CS (prosp)	DGS vs. FH	478	714	Max and mand	NS	X-Guide, X-Nav Technologies	NR
Ting-Mao Sun et al. [35] (2020)	CS (prosp)	DGS vs. FH	NS	96	Max and mand	Flapless	AQNavi, (TITC Ltd., Kaohsiung, Taiwan)	NR
Stefanelli et al. [38] (2020)	CS (prosp)	DGS	13	77	Max and mand	NS	Navident (ClaroNav Inc., Toronto, ON, Canada)	4 implants lost through lack of osseointegration.
Du-Hyeong Lee et al. [39] (2016)	CS (prosp)	SGS	11	21	Max and mand	Flapless	R2GATE 1.0; MegaGen Implant, Gyeongbuk, Korea	NR
Oguz Ozan et al. [50] (2009)	CS (prosp)	SGS	30	30	NR	Flapless	Stent Cad; Media Lab Software, La Spezia, Italy	NR
Van Assche et al. [51] (2010)	CS (prosp)	SGS	8	21	Max and mand	Flapless	Procera (Nobel Biocare AB, Göteburg, Sweden)	NR
Platzer et al. [52] (2011)	CS (prosp)	SGS	5	15	Mand	Flapless	Simplant Materialise Dental, Leuven, Belgium	NR
Ahmet Ersan Ersoy et al. [40] (2008)	CS (prosp)	SGS	14	29	Max and mand	Flap and flapless	Stent Cad, Media Lab Software, La Spezia, Italy	NR
Dong Wu et al. [46] (2020)	CS (retrosp)	DGS vs. SGS	54	95	NR	Flap	DHC-DI3E, Suzhou Digital-health Care Co., Ltd., China	NR
Stefanelli et al. [53]	CS (retrosp)	DGS	59	136	Max and mand	NS	Navident (ClaroNav Inc., Toronto, ON, Canada)	NR
Stefanelli et al. [54]	CS (retrosp)	DGS	14	56	Max	NS	Navident (ClaroNav Inc., Toronto, ON, Canada)	
Fürhauser et al. [55]	CS (retrosp)	SGS	27	27	Max	Flapless	NobelClinician (Nobel Biocare, Gothenburg, Sweden)	NR

Note. RCT: randomized clinical trial; CS: case series; prosp: prospective; retrosp: retrospective; max: maxilla; mand: mandibular; NS: not specified; NR: not reported.

**Table 2 dentistry-11-00150-t002:** Accuracy values in the included studies.

	Dynamic Guided				Static Guided				Freehand			
Study	Platform Deviation (SD)	Angular Deviation (SD)	Apical Deviation (SD)	Vertical Deviation (SD)	Platform Deviation (SD)	Angular Deviation (SD)	Apical Deviation (SD)	Vertical Deviation (SD)	Platform Deviation (SD)	Angular Deviation (SD)	Apical Deviation (SD)	Vertical Deviation (SD)
Dechawat Kaewsiri et al. [42] (2019)	1.05 (0.44)	3.06 (1.37)	1.29 (0.50)	NA	0.97 (0.44)	2.84 (1.71)	1.28 (0.46)	NA	NA	NA	NA	NA
Paweena Yimarj et al. [47] (2020)	1.24 (0.39)	3.78 (1.84)	1.58 (0.56)	NA	1.04 (0.67)	4.08 (1.69)	1.54 (0.79)	NA	NA	NA	NA	NA
Aktolun Aydemir et al. [37] (2020)	1.01 (0.07)	5.59 (0.39)	1.83 (0.12)	NA	NA	NA	NA	NA	1.70 (0.13)	10.04 (0.83)	2.51 (0.21)	NA
Palita Smitkarn et al. [45] (2019)	NA	NA	NA	NA	0.9 (0.8)	2.8 (2.6)	1.2 (0.9)	NA	1.3 (0.7)	7.0 (7.0)	2.2 (1.2)	NA
Faris Younes et al. [41] (2018)	NA	NA	NA	NA	0.94 (0.1)	4.25 (0.89)	1.22 (0.18)	NA	1.45 (0.1)	6.99 (0.87)	2.11 (0.18)	NA
Pei Shen et al. [48] (2015)	NA	NA	NA	NA	1.18 (0.72)	4.21 (1.91)	1.43 (0.74)	0.54 (0.29)	2.07 (0.51)	8.84 (4.64)	2.89 (1.02)	0.78 (0.33)
Farley et al. [36] (2013)	NA	NA	NA	NA	0.64 (0.37)	4.26 (NA)	1.11 (0.71)	(−) 1.20 (0.70)	1.15 (0.57)	7.14 (NA)	1.84 (0.97)	(−) 1.51 (1.02)
Magrin et al. [43] (2019)	NA	NA	NA	NA	2.34 (1.01)	2.2 (1.1)	2.53 (1.11)	NA	1.93 (0.95)	3.5 (1.6)	2.19 (1.00)	NA
Pellegrino et al. [44] (2017)	0.78 (0.20)	NA	1.04 (0.29)	NA	NA	NA	NA	NA	NA	NA	NA	NA
Elian et al. [49] (2008)	0.89 (0.53)	3.78 (2.76)	0.96 (0.50)	NA	NA	NA	NA	NA	NA	NA	NA	NA
Block et al. [21] (2017)	1.25 (0.65)	3.26 (2.24)	1.43 (0.73)	0.84 (0.68)	NA	NA	NA	NA	1.78 (0.77)	6.50 (4.21)	2.27 (1.02)	1.12 (0.83)
Ting-Mao Sun et al. [35] (2020)	0.73 (0.13)	3.24 (0.36)	NA	NA	1.00 (0.15)	4.54 (0.29)	NA	NA	1.42 (0.25)	6.12 (0.12)	NA	NA
Stefanelli et al. [38] (2020)	0.66 (0.32)	2.7 (0.99)	1 (0.35)	0.57 (0.29)	NA	NA	NA	NA	NA	NA	NA	NA
Du-Hyeong Lee et al. [39] (2016)	NA	NA	NA	NA	0.64 (0.29)	2.21 (1.04)	NA	0.93 (0.38)	NA	NA	NA	NA
Oguz Ozan et al. [50] (2009)	NA	NA	NA	NA	O.87 (0.4)	2.91 (1.3)	0.95 (0.6)	NA	NA	NA	NA	NA
Van Assche et al. [51] (2010)	NA	NA	NA	NA	0.7 (0.34)	2.7 (1.9)	1.0 (0.7)	NA	NA	NA	NA	NA
Platzer et al. [52] (2011)	NA	NA	NA	NA	0.27 (0.19)	14 (11.6)	0.15 (0.13)	0.28 (0.19)	NA	NA	NA	NA
Ahmet Ersan Ersoy et al. [40] (2008)	NA	NA	NA	NA	1.08 (0.6)	4.45 (1.64)	1.3 (0.64)	NA	NA	NA	NA	NA
Dong Wu et al. [43] (2020)	1.36 (0.65)	3.71 (1.32)	1.48 (0.65)	NA	1.22 (0.70)	4.34 (2.22)	1.33 (0.73)	NA	NA	NA	NA	NA
Luigi V. Stefanelli et al. [53] (2020)	0.67 (0.29)	2.5 (1.04)	0.99 (0.33)	0.55 (0.25)	NA	NA	NA	NA	NA	NA	NA	NA
Stefanelli et al. [54] (2020)	0.64 (0.37)	2.49 (1.14)	0.89 (0.42)	0.46 (0.26)	NA	NA	NA	NA	NA	NA	NA	NA
Fürhauser et al. [55] (2014)	NA	NA	NA	NA	0.84 (0.44)	2.7 (2.6)	1.16 (0.69)	0.52 (0.39)	NA	NA	NA	NA

All measurements are expressed in mm except angular deviation, which is in degrees. NA, not available.

**Table 3 dentistry-11-00150-t003:** Comparison of platform deviation between DGS and SGS, and between DGS and FH.

Study	Mean (SD)		
	Dynamic	Static	Mean Difference, % (CI)
DGS vs. SGS			
Ting-Mao Sun et al. [35] (2020)	0.73 (0.13)	1.00 (0.15)	−0.27 (−0.34, −0.20)
Dechawat Kaewsiri et al. [42] (2019)	1.05 (0.44)	0.97 (0.44)	0.08 (−0.23, 0.39)
Paweena Yimarj et al. [47] (2020)	1.24 (0.39)	1.04 (0.67)	0.20 (−0.08, 0.48)
Dong Wu et al. [46] (2020)	1.36 (0.65)	1.22 (0.70)	0.14 (−0.14, 0.42)
Model for all studies			0.02 (−0.27, 0.31)
**DGS vs. freehand**	Dynamic	Freehand	
Block et al. [21] (2017)	1.25 (0.65)	1.78 (0.77)	−0.53 (−0.68, −0.38)
Ting-Mao Sun et al. [35] (2020)	0.73 (0.13)	1.42 (0.25)	−0.69 (−0.79, −0.59)
Aktolun Aydemir et al. [37] (2020)	1.01 (0.07)	1.70 (0.13)	−0.69 (−0.73, −0.65)
Model for all studies			−0.66 (−0.74, −0.59)

**Table 4 dentistry-11-00150-t004:** Comparison of the angular deviation between DGS and SGS and between DGS and FH.

Study	Mean (SD)		
	Dynamic	Static	Mean Difference, % (CI)
DGS vs. SGS			
Ting-Mao Sun et al. [35] (2020)	3.24 (0.36)	4.54 (0.29)	−1.30 (−1.46, −1.14)
Dechawat Kaewsiri et al. [42] (2019)	3.06 (1.37)	2.84 (1.71)	0.22 (−0.89, 1.33)
Paweena Yimarj et al. [47] (2020)	3.78 (1.84)	4.08 (1.69)	−0.30 (−1.19, 0.59)
Dong Wu et al. [46] (2020)	3.71 (1.32)	4.34 (2.22)	−0.63 (−1.34, 0.08)
Model for all studies			−0.62 (−1.33, 0.09)
**DGS vs. freehand**	Dynamic	Freehand	
Block et al. [21] (2017)	3.26 (2.24)	6.5 (4.21)	−3.24 (−4.01, −2.47)
Ting-Mao Sun et al. [35] (2020)	3.24 (0.36)	6.12 (0.12)	−2.88 (−3.01, −2.75)
Aktolun Aydemir et al. [37] (2020)	5.59 (0.39)	10.04 (0.83)	−4.41 (−4.68, −4.14)
Model for all studies			−3.52 (−4.69, −2.35)

**Table 5 dentistry-11-00150-t005:** Comparison of the apical deviation between DGS and SGS and between DGS and FH.

Study	Mean (SD)		
	Dynamic	Static	Mean Difference, % (CI)
DGS vs. SGS			
Dechawat Kaewsiri et al. [42] (2019)	1.29 (0.50)	1.28 (0.46)	0.01 (−0.33, 0.35)
Paweena Yimarj et al. [47] (2020)	1.58 (0.56)	1.54 (0.79)	0.04 (−0.31, 0.39)
Dong Wu et al. [46] (2020)	1.48 (0.65)	1.33 (0.73)	0.15 (−0.13, 0.43)
Model for all studies			0.08 (−0.11, 0.26)
**DGS vs. freehand**	Dynamic	Freehand	
Block et al. [21] (2017)	1.43 (0.73)	2.27 (1.02)	−0.84 (−1.03, −0.65)
Aktolun Aydemir et al. [37] (2020)	1.83 (0.12)	2.51 (0.21)	−0.68 (−0.75, −0.61)
Model for all studies			−0.73 (−0.88, −0.59)

## Data Availability

The data presented in this study are available in the included studies of this systematic review.

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
