# Peer review of "Dynamic Implant Surgery—An Accurate Alternative to Stereolithographic Guides—Systematic Review and Meta-Analysis"

_dentistry, 2023, doi:10.3390/dj11060150_

Round 1
Reviewer 1 Report
This Systematic review and meta-analysis is interesting, it has an appropriate and detailed design the results obtained reflect the applied methodology supported by the statistical analysis of easy interpretation and answers the focus question (PICO).
Author Response
Response to Reviewer 1 Comments
This Systematic review and meta-analysis is interesting, it has an appropriate and detailed design the results obtained reflect the applied methodology supported by the statistical analysis of easy interpretation and answers the focus question (PICO).
Thank you very much for your comments.
In reference to the English language and style the paper has been summited to an extensive correction.

Reviewer 2 Report
Major concerns:
1. Extensive english and style editing is required
2. A comparison between SGS and free hand is missing and required
Minor Concerns:
1. Please be consistent with the term you are using, especially the four different paramters you evaluated which you slightly change their names along the manuscript
2. In Table 1 you cite the article of Aydemir et al. which evaluated DGS ns FH and in the surgical complications you write 'No guide stabilitiy', however according to the techniques evaluated in the study there was no guide at all
3. Figure legends in all figures are lacking and the figures are hard to undestand. please revise
Author Response
Response to Reviewer 2 Comments
Major concerns:
- Extensive English and style editing is required.
Response to major concern point 1:
The manuscript has been reviewed and corrected by a native English-speaking colleague with experience in correcting manuscripts.
- A comparison between SGS and free hand is missing and required.
Response to major concern point 2:
A direct comparison between the SGS and free hand has not been done because the following different reasons:
Firstly, the DGS is a newer technique than the SGS and the free hand method. Consequently, this technique has less literature support than its alternatives. Since the application of the DGS in the dental implantology field is to place implants more accurately the aim of this study was to compare the Dynamic guided method to its alternative guidance systems (SGS and free hand) rather than a comparation of the alternative methods between them.
Secondly, the study would have been excessively extensive if an additional comparison between SGS and free hand would have been included.
Nevertheless, the study includes forest plot for all included studies that reports data (DGS, SGS, FH) in each different parameter (platform, apical, angular and vertical). Despite there isn’t a direct statistical comparison between them the forest plots offers a visual idea of the accuracy of each technique.
Minor Concerns:
- Please be consistent with the term you are using, especially the four different parameters you evaluated which you slightly change their names along the manuscript.
Response to minor concern point 1:
All the different terms used for the evaluated parameters have been unified to a single term for each one of the parameters. Also, the definitions of these terms have been revised and improved.
- In Table 1 you cite the article of Aydemir et al. which evaluated DGS ns FH and in the surgical complications you write 'No guide stabilitiy', however according to the techniques evaluated in the study there was no guide at all.
Response to minor concern point 2:
Thank you very much for the observation. Correct, there wasn’t any surgical guide but there was a radiopaque stent to sustain the fiducial tags. The author of the mentioned article presents in his results that the retention of this radiopaque stent wasn’t satisfactory in two patients. Because the fiducial marker is an essential part for the dynamic guided system, we consider that it is still important to be mentioned in the table. That’s why it has been corrected and reworded to specify that it was the radiopaque stent what had lack of stability instead of the non-existing surgical guide.
- Figure legends in all figures are lacking and the figures are hard to undestand. please revise.
Response to minor concern point 3:
All the figure legends have been revised and reworded to have better understanding.

Reviewer 3 Report
Dear authors,
I evaluated the article titled “Dynamic Implant Navigation Surgery, a Valid Alternative Guidance Method. Systematic Review and Meta-analysis”. The aim of this research was to compare the accuracy between the dynamic guided surgery (DGS) and alternative implant guidance methods, SGS and free hand.
The article is interesting but many concerns were raised.
———-----------
Title: It must be improved
ABSTRACT: it is incomplete. See M&M used (it is poorly described)
INTRO: I considered enough and well done.
M&M:
There is a fail in the PICO question. please, review it.
Could the authors justify why included only 2 databases? I suggested including more databases due to the relevance of the topic
The authors reported results in different section => Figure 1 - there is unnecessary characters. Remove them
- Please, adjust or clarify the limit of time for the screening.
- Exclusion criteria can be considered for lack of content.
RESULTS
Where is the result for “manual search” and “Grey literature from the Google Scholar Beta database was also searched to retrieve studies published in journals not indexed in the major databases”? This results were not inserted in the article. Include them.
Figure 5 - it was not possible to evaluate.
Meta-analysis: interesting. But, it is necessary to review the figures and part of the result. there are extremes; moreover, the direct comparison was limited to few studies. Try to explain it.
- Quality of assessment were well done.
- Review the conclusion
Author Response
Response to Reviewer 3 Comments
Title: It must be improved
Response:
A modification of the title has been done as suggested.
ABSTRACT: it is incomplete. See M&M used (it is poorly described).
More information has been included in M&M.
INTRO: I considered enough and well done.
Response:
Thank you.
M&M:
There is a fail in the PICO question. please, review it.
Response:
PICO question has been reviewed.
Could the authors justify why included only 2 databases? I suggested to include more databases due to the relevance of the topic.
Response:
Thank you for your suggestions. According to your recommendation regarding the importance of this topic, not only 2 electronic databases were scrutinized, but also the authors performed a systematic review which included different both an electronic database (Cochrane, MEDLINE via PuBbMed, google scholar and a manual search Scopus from certain indexed dental implant journals., in order to obtain the major relevant papers for this study (M&M>Search Strategy (page 3 and 4).
The authors reported results in different section => Figure 1 - there is unnecessary characters. Remove them.
Response:
It has been removed.
- Please, adjust or clarify the limit of time for the screening.
Response:
Limit of time for screening has been clarified.
- Exclusion criteria can be considered with lack of content.
Response:
Thank you for your comment. All the exclusion criteria used is mentioned in study selection. A slight modification trying to be more precise has been done but no additional criteria was used.
RESULTS
Where is the result for “manual search” and “Grey literature from the Google Scholar Beta database was also searched to retrieve studies published in journals not indexed in the major databases”? These results were not inserted in the article. Include them.
Response:
Thank you for your suggestion. According to your recommendation, these has been added at the Prisma flowchart (Manual search and google scholar). The results of this search has been included.
Figure 5 - it was not possible to evaluate.
Response:
Thank you for your comment. I realized that part of the Fig. 5 was laterally moved outside of the “frame” and part of the data were not visible. It has been corrected.
Meta-analysis: interesting. But it is necessary to review the figures and part of the result. there are extremes; moreover, the direct comparison was limited to few studies. Try to explain it.
Response:
The direct comparison was limited to a few studies because we were analysing four parameters and not all the studies were analysing the four of them. Consequently, these studies couldn’t be included for all the analyses. Also, the number of studies available in the literature meeting the inclusion criteria are not very numerous.
The technique is described as very technical-sensitive and the results can be affected for multiple variables reason that explains the extremes or heterogeneity. (Extensively explained in the Discussion section when the study limitations are analysed).
- Quality of assessment were well done.
Response:
Thank you.
- Review the conclusion
Response:
Conclusion has been reviewed.

Round 2
Reviewer 2 Report
no further comments
Author Response
no further comments
Response:
Thank you very much.
Reviewer 3 Report
Dear authors, thank you for all responses. But I double-checked the article and where are all modifications?
- Figure 1 is a results and it is in the M&M section. The authors are responding and were not effecting the modifications.
- clarify the databases findings... where is the data found in the Cochrane, MEDLINE via PubMed, google scholar and a manual search Scopus from certain indexed dental implant journals??? This part is extremely important for the reproducibility of the study. The authors are responding incorrectly, comparing with the article.
- Please, double-check the meta-analysis. I did recommendations and no modification was observed. Please, send it to a person/professional that understanding of statistical.
Author Response
Dear authors, thank you for all responses. But I double-checked the article and where are all modification:
Response:
Dear reviewer please apologies for the inconvenience. The corrections were done in two different version and the document that was uploaded contained only part of your corrections.
- Figure 1 is a result and it is in the M&M section. The authors are responding and were not affecting the modifications.
Response:
Thank you for your comment. Reference and figure have been moved to results.
- clarify the databases findings... where is the data found in the Cochrane, MEDLINE via PubMed, google scholar and a manual search Scopus from certain indexed dental implant journals??? This part is extremely important for the reproducibility of the study. The authors are responding incorrectly, comparing with the article.
Response:
Thank you for your suggestion. The data collection has been reviewed and the articles found in each database has been specified in the PRISMA flowchart figure.
- Please, double-check the meta-analysis. I did recommendations and no modification was observed. Please, send it to a person/professional that understanding of statistical.
Response:
Thank you very much for your corrections. I am happy to confirm that the statistical analysis has been done by a professional statistic and the data has been double checked. Agreeing with you, there are extremes. Direct consequences of the study variability (Techniques described as a very technical-sensitive, the size of the patient sample, variability of the patient characteristics). That is why it has been suggested to interpretate the data with caution and the need for more standardized RCT.
Despite that the direct comparison was limited to a few studies. The available data offered enough information for obtaining a meta-analysis in all the parameters with the exemption of the vertical deviation.

Round 3
Reviewer 3 Report
Congratulations!
Author Response
Thank you very much
